# Value Prediction Network

**Junhyuk Oh**[†]     **Satinder Singh**[†]     **Honglak Lee**[∗,†]
†University of Michigan
∗Google Brain
{junhyuk,baveja,honglak}@umich.edu, honglak@google.com

## Abstract

This paper proposes a novel deep reinforcement learning (RL) architecture, called Value Prediction Network (VPN), which integrates model-free and model-based RL methods into a single neural network. In contrast to typical model-based RL methods, VPN learns a dynamics model whose abstract states are trained to make option-conditional predictions of future values (discounted sum of rewards) rather than of future observations. Our experimental results show that VPN has several advantages over both model-free and model-based baselines in a stochastic environment where careful planning is required but building an accurate observation-prediction model is difficult. Furthermore, VPN outperforms Deep Q-Network (DQN) on several Atari games even with short-lookahead planning, demonstrating its potential as a new way of learning a good state representation.

## 1   Introduction

Model-based reinforcement learning (RL) approaches attempt to learn a model that predicts future observations conditioned on actions and can thus be used to simulate the real environment and do multi-step lookaheads for planning. We will call such models an *observation-prediction model* to distinguish it from another form of model introduced in this paper. Building an accurate observation-prediction model is often very challenging when the observation space is large [23, 5, 13, 4] (e.g., high-dimensional pixel-level image frames), and even more difficult when the environment is stochastic. Therefore, a natural question is whether it is possible to plan without predicting future observations.

In fact, raw observations may contain information unnecessary for planning, such as dynamically changing backgrounds in visual observations that are irrelevant to their value/utility. The starting point of this work is the premise that what planning truly requires is the ability to predict the rewards and values of future states. An observation-prediction model relies on its predictions of observations to predict future rewards and values. What if we could predict future rewards and values directly without predicting future observations? Such a model could be more easily learnable for complex domains or more flexible for dealing with stochasticity. In this paper, we address the problem of learning and planning from a *value-prediction model* that can directly generate/predict the value/reward of future states without generating future observations.

Our main contribution is a novel neural network architecture we call the *Value Prediction Network* (VPN). The VPN combines model-based RL (i.e., learning the dynamics of an abstract state space sufficient for computing future rewards and values) and model-free RL (i.e., mapping the learned abstract states to rewards and values) in a unified framework. In order to train a VPN, we propose a combination of *temporal-difference search* [28] (TD search) and n-step Q-learning [20]. In brief, VPNs learn to predict values via Q-learning and rewards via supervised learning. At the same time, VPNs perform lookahead planning to choose actions and compute bootstrapped target Q-values.

Our empirical results on a 2D navigation task demonstrate the advantage of VPN over model-free baselines (e.g., Deep Q-Network [21]). We also show that VPN is more robust to stochasticity in the environment than an observation-prediction model approach. Furthermore, we show that our VPN outperforms DQN on several Atari games [2] even with short-lookahead planning, which suggests

that our approach can be potentially useful for learning better abstract-state representations and reducing sample-complexity.

## 2    Related Work

**Model-based Reinforcement Learning.**    Dyna-Q [32, 34, 39] integrates model-free and model-based RL by learning an observation-prediction model and using it to generate samples for Q-learning in addition to the model-free samples obtained by acting in the real environment. Gu et al. [7] extended these ideas to continuous control problems. Our work is similar to Dyna-Q in the sense that planning and learning are integrated into one architecture. However, VPNs perform a lookahead tree search to choose actions and compute bootstrapped targets, whereas Dyna-Q uses a learned model to generate imaginary samples. In addition, Dyna-Q learns a model of the environment separately from a value function approximator. In contrast, the dynamics model in VPN is combined with the value function approximator in a single neural network and indirectly learned from reward and value predictions through backpropagation.

Another line of work [23, 4, 8, 30] uses observation-prediction models not for planning, but for improving exploration. A key distinction from these prior works is that our method learns abstract-state dynamics not to predict future observations, but instead to predict future rewards/values. For continuous control problems, deep learning has been combined with model predictive control (MPC) [6, 18, 26], a specific way of using an observation-prediction model. In cases where the observation-prediction model is differentiable with respect to continuous actions, backpropagation can be used to find the optimal action [19] or to compute value gradients [11]. In contrast, our work focuses on learning and planning using lookahead for discrete control problems.

Our VPNs are related to Value Iteration Networks [35] (VINs) which perform value iteration (VI) by approximating the Bellman-update through a convolutional neural network (CNN). However, VINs perform VI over the entire state space, which in practice requires that 1) the state space is small and representable as a vector with each dimension corresponding to a separate state and 2) the states have a topology with local transition dynamics (e.g., 2D grid). VPNs do not have these limitations and are thus more generally applicable, as we will show empirically in this paper.

VPN is close to and in-part inspired by Predictron [29] in that a recurrent neural network (RNN) acts as a transition function over abstract states. VPN can be viewed as a *grounded* Predictron in that each rollout corresponds to the transition in the environment, whereas each rollout in Predictron is purely abstract. In addition, Predictrons are limited to uncontrolled settings and thus policy evaluation, whereas our VPNs can learn an optimal policy in controlled settings.

**Model-free Deep Reinforcement Learning.**    Mnih et al. [21] proposed the Deep Q-Network (DQN) architecture which learns to estimate Q-values using deep neural networks. A lot of variations of DQN have been proposed for learning better state representation [37, 16, 9, 22, 36, 24], including the use of memory-based networks for handling partial observability [9, 22, 24], estimating both state-values and advantage-values as a decomposition of Q-values [37], learning successor state representations [16], and learning several auxiliary predictions in addition to the main RL values [12]. Our VPN can be viewed as a model-free architecture which 1) decomposes Q-value into reward, discount, and the value of the next state and 2) uses multi-step reward/value predictions as auxiliary tasks to learn a good representation. A key difference from the prior work listed above is that our VPN learns to simulate the future rewards/values which enables planning. Although STRAW [36] can maintain a sequence of future actions using an external memory, it cannot explicitly perform planning by simulating future rewards/values.

**Monte-Carlo Planning.**    Monte-Carlo Tree Search (MCTS) methods [15, 3] have been used for complex search problems, such as the game of Go, where a simulator of the environment is already available and thus does not have to be learned. Most recently, *AlphaGo* [27] introduced a *value network* that directly estimates the value of state in Go in order to better approximate the value of leaf-node states during tree search. Our VPN takes a similar approach by predicting the value of abstract future states during tree search using a value function approximator. *Temporal-difference search* [28] (TD search) combined TD-learning with MCTS by computing target values for a value function approximator through MCTS. Our algorithm for training VPN can be viewed as an instance of TD search, but it learns the dynamics of future rewards/values instead of being given a simulator.

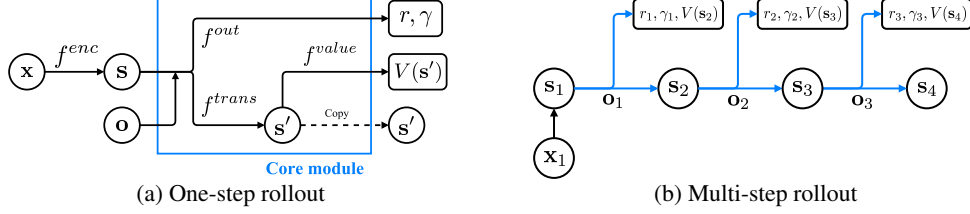

(a) One-step rollout                                      (b) Multi-step rollout

Figure 1: Value prediction network. (a) VPN learns to predict immediate reward, discount, and the value of the next abstract-state. (b) VPN unrolls the core module in the abstract-state space to compute multi-step rollouts.

# 3   Value Prediction Network

The value prediction network is developed for semi-Markov decision processes (SMDPs). Let $\mathbf{x}_t$ be the observation or a history of observations for partially observable MDPs (henceforth referred to as just observation) and let $\mathbf{o}_t$ be the *option* [33, 31, 25] at time $t$. Each option maps observations to primitive actions, and the following Bellman equation holds for all policies $\pi$: $Q^{\pi}(\mathbf{x}_t, \mathbf{o}_t) = \mathbb{E}[\sum_{i=0}^{k-1} \gamma^i r_{t+i} + \gamma^k V^{\pi}(\mathbf{x}_{t+k})]$, where $\gamma$ is a discount factor, $r_t$ is the immediate reward at time $t$, and $k$ is the number of time steps taken by the option $\mathbf{o}_t$ before terminating in observation $\mathbf{x}_{t+k}$.

A VPN not only learns an option-value function $Q_{\theta}(\mathbf{x}_t, \mathbf{o}_t)$ through a neural network parameterized by $\theta$ like model-free RL, but also learns the dynamics of the rewards/values to perform planning. We describe the architecture of VPN in Section 3.1. In Section 3.2, we describe how to perform planning using VPN. Section 3.3 describes how to train VPN in a Q-Learning-like framework [38].

## 3.1   Architecture

The VPN consists of the following modules parameterized by $\theta = \{\theta^{enc}, \theta^{value}, \theta^{out}, \theta^{trans}\}$:

$$\textbf{Encoding } f_{\theta}^{enc} : \mathbf{x} \mapsto \mathbf{s} \qquad\qquad \textbf{Value } f_{\theta}^{value} : \mathbf{s} \mapsto V_{\theta}(\mathbf{s})$$
$$\textbf{Outcome } f_{\theta}^{out} : \mathbf{s}, \mathbf{o} \mapsto r, \gamma \qquad\qquad \textbf{Transition } f_{\theta}^{trans} : \mathbf{s}, \mathbf{o} \mapsto \mathbf{s}'$$

- **Encoding** module maps the observation ($\mathbf{x}$) to the abstract state ($\mathbf{s} \in \mathbb{R}^m$) using neural networks (e.g., CNN for visual observations). Thus, $\mathbf{s}$ is an *abstract-state representation which will be learned by the network (and not an environment state or even an approximation to one).*

- **Value** module estimates the value of the abstract-state ($V_{\theta}(\mathbf{s})$). Note that the value module is not a function of the observation, but a function of the abstract-state.

- **Outcome** module predicts the option-reward ($r \in \mathbb{R}$) for executing the option $\mathbf{o}$ at abstract-state $\mathbf{s}$. If the option takes $k$ primitive actions before termination, the outcome module should predict the discounted sum of the $k$ immediate rewards as a scalar. The outcome module also predicts the option-discount ($\gamma \in \mathbb{R}$) induced by the number of steps taken by the option.

- **Transition** module transforms the abstract-state to the next abstract-state ($\mathbf{s}' \in \mathbb{R}^m$) in an option-conditional manner.

Figure 1a illustrates the *core* module which performs 1-step rollout by composing the above modules: $f_{\theta}^{core} : \mathbf{s}, \mathbf{o} \mapsto r, \gamma, V_{\theta}(\mathbf{s}'), \mathbf{s}'$. The core module takes an abstract-state and option as input and makes separate option-conditional predictions of the option-reward (henceforth, reward), the option-discount (henceforth, discount), and the value of the abstract-state at option-termination. By combining the predictions, we can estimate the Q-value as follows: $Q_{\theta}(\mathbf{s}, \mathbf{o}) = r + \gamma V_{\theta}(\mathbf{s}')$. In addition, the VPN recursively applies the core module to predict the sequence of future abstract-states as well as rewards and discounts given an initial abstract-state and a sequence of options as illustrated in Figure 1b.

## 3.2   Planning

VPN has the ability to simulate the future and plan based on the simulated future abstract-states. Although many existing planning methods (e.g., MCTS) can be applied to the VPN, we implement a simple planning method which performs rollouts using the VPN up to a certain depth (say $d$), henceforth denoted as *planning depth*, and aggregates all intermediate value estimates as described in Algorithm 1 and Figure 2. More formally, given an abstract-state $\mathbf{s} = f_{\theta}^{enc}(\mathbf{x})$ and an option $\mathbf{o}$, the

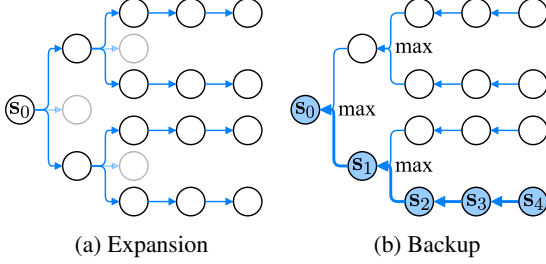

|   |   |
|---|---|
| (a) Expansion | (b) Backup |

Figure 2: Planning with VPN. (a) Simulate $b$-best options up to a certain depth ($b = 2$ in this example). (b) Aggregate all possible returns along the best sequence of future options.

---

**Algorithm 1** Q-value from $d$-step planning
---

**function** Q-PLAN($\mathbf{s}, \mathbf{o}, d$)
    $r, \gamma, V(\mathbf{s}'), \mathbf{s}' \leftarrow f_\theta^{core}(\mathbf{s}, \mathbf{o})$
    **if** $d = 1$ **then**
        **return** $r + \gamma V(\mathbf{s}')$
    **end if**
    $\mathcal{A} \leftarrow b$-best options based on $Q^1(\mathbf{s}', \mathbf{o}')$
    **for** $\mathbf{o}' \in \mathcal{A}$ **do**
        $q_{\mathbf{o}'} \leftarrow$ Q-PLAN($\mathbf{s}', \mathbf{o}', d - 1$)
    **end for**
    **return** $r + \gamma \left[ \frac{1}{d} V(\mathbf{s}') + \frac{d-1}{d} \max_{\mathbf{o}' \in \mathcal{A}} q_{\mathbf{o}'} \right]$
**end function**

---

Q-value calculated from $d$-step planning is defined as:

$$Q_\theta^d(\mathbf{s}, \mathbf{o}) = r + \gamma V_\theta^d(\mathbf{s}') \qquad V_\theta^d(\mathbf{s}) = \begin{cases} V_\theta(\mathbf{s}) & \text{if } d = 1 \\ \frac{1}{d} V_\theta(\mathbf{s}) + \frac{d-1}{d} \max_{\mathbf{o}} Q_\theta^{d-1}(\mathbf{s}, \mathbf{o}) & \text{if } d > 1, \end{cases} \qquad (1)$$

where $\mathbf{s}' = f_\theta^{trans}(\mathbf{s}, \mathbf{o})$, $V_\theta(\mathbf{s}) = f_\theta^{value}(\mathbf{s})$, and $r, \gamma = f_\theta^{out}(\mathbf{s}, \mathbf{o})$. Our planning algorithm is divided into two steps: expansion and backup. At the expansion step (see Figure 2a), we recursively simulate options up to a depth of $d$ by unrolling the core module. At the backup step, we compute the weighted average of the direct value estimate $V_\theta(\mathbf{s})$ and $\max_{\mathbf{o}} Q_\theta^{d-1}(\mathbf{s}, \mathbf{o})$ to compute $V_\theta^d(\mathbf{s})$ (i.e., value from $d$-step planning) in Equation 1. Note that $\max_{\mathbf{o}} Q_\theta^{d-1}(\mathbf{s}, \mathbf{o})$ is the average over $d - 1$ possible value estimates. We propose to compute the uniform average over all possible returns by using weights proportional to 1 and $d - 1$ for $V_\theta(\mathbf{s})$ and $\max_{\mathbf{o}} Q_\theta^{d-1}(\mathbf{s}, \mathbf{o})$ respectively. Thus, $V_\theta^d(\mathbf{s})$ is the uniform average of $d$ expected returns along the path of the best sequence of options as illustrated in Figure 2b.

To reduce the computational cost, we simulate only $b$-best options at each expansion step based on $Q^1(\mathbf{s}, \mathbf{o})$. We also find that choosing only the best option after a certain depth does not compromise the performance much, which is analogous to using a default policy in MCTS beyond a certain depth. This heuristic visits reasonably good abstract states during planning, though a more principled way such as UCT [15] can also be used to balance exploration and exploitation. This planning method is used for choosing options and computing target Q-values during training, as described in the following section.

### 3.3 Learning

VPN can be trained through any existing value-based RL algorithm for the value predictions combined with supervised learning for reward and discount predictions. In this paper, we present a modification of n-step Q-learning [20] and TD search [28]. The main idea is to generate trajectories by following $\epsilon$-greedy policy based on the planning method described in Section 3.2. Given an n-step trajectory $\mathbf{x}_1, \mathbf{o}_1, r_1, \gamma_1, \mathbf{x}_2, \mathbf{o}_2, r_2, \gamma_2, ..., \mathbf{x}_{n+1}$ generated by the $\epsilon$-greedy policy, $k$-step predictions are defined as follows:

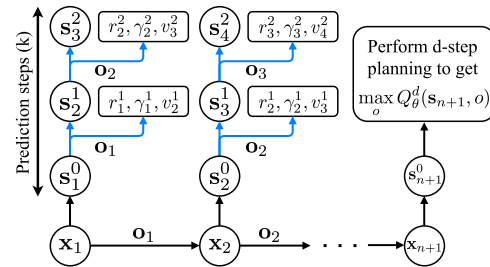

Figure 3: Illustration of learning process.

$$\mathbf{s}_t^k = \begin{cases} f_\theta^{enc}(\mathbf{x}_t) & \text{if } k = 0 \\ f_\theta^{trans}(\mathbf{s}_{t-1}^{k-1}, \mathbf{o}_{t-1}) & \text{if } k > 0 \end{cases} \qquad v_t^k = f_\theta^{value}(\mathbf{s}_t^k) \qquad r_t^k, \gamma_t^k = f_\theta^{out}(\mathbf{s}_t^{k-1}, \mathbf{o}_t).$$

Intuitively, $\mathbf{s}_t^k$ is the VPN's $k$-step prediction of the abstract-state at time $t$ predicted from $\mathbf{x}_{t-k}$ following options $\mathbf{o}_{t-k}, ..., \mathbf{o}_{t-1}$ in the trajectory as illustrated in Figure 3. By applying the value and the outcome module, VPN can compute the $k$-step prediction of the value, the reward, and the discount. The $k$-step prediction loss at step $t$ is defined as:

$$\mathcal{L}_t = \sum_{l=1}^{k} \left( R_t - v_t^l \right)^2 + \left( r_t - r_t^l \right)^2 + \left( \log_\gamma \gamma_t - \log_\gamma \gamma_t^l \right)^2$$

where $R_t = \begin{cases} r_t + \gamma_t R_{t+1} & \text{if } t \leq n \\ \max_{\mathbf{o}} Q_{\theta^-}^d(\mathbf{s}_{n+1}, \mathbf{o}) & \text{if } t = n+1 \end{cases}$ is the target value, and $Q_{\theta^-}^d(\mathbf{s}_{n+1}, o)$ is the Q-value computed by the $d$-step planning method described in 3.2. Intuitively, $\mathcal{L}_t$ accumulates losses over 1-step to $k$-step predictions of values, rewards, and discounts. We find that applying $\log_\gamma$ for the discount prediction loss helps optimization, which amounts to computing the squared loss with respect to the number of steps.

Our learning algorithm introduces two hyperparameters: the number of prediction steps ($k$) and planning depth ($d_{train}$) used for choosing options and computing bootstrapped targets. We also make use of a *target network* parameterized by $\theta^-$ which is synchronized with $\theta$ after a certain number of steps to stabilize training as suggested by [20]. The loss is accumulated over n-steps and the parameter is updated by computing its gradient as follows: $\nabla_\theta \mathcal{L} = \sum_{t=1}^n \nabla_\theta \mathcal{L}_t$. The full algorithm is described in the supplementary material.

### 3.4 Relationship to Existing Approaches

VPN is model-based in the sense that it learns an abstract-state transition function sufficient to predict rewards/discount/values. Meanwhile, VPN can also be viewed as model-free in the sense that it learns to directly estimate the value of the abstract-state. From this perspective, VPN exploits several auxiliary prediction tasks, such as reward and discount predictions to learn a good abstract-state representation. An interesting property of VPN is that its planning ability is used to compute the bootstrapped target as well as choose options during Q-learning. Therefore, as VPN improves the quality of its future predictions, it can not only perform better during evaluation through its improved planning ability, but also generate more accurate target Q-values during training, which encourages faster convergence compared to conventional Q-learning.

## 4 Experiments

Our experiments investigated the following questions: 1) Does VPN outperform model-free baselines (e.g., DQN)? 2) What is the advantage of planning with a VPN over observation-based planning? 3) Is VPN useful for complex domains with high-dimensional sensory inputs, such as Atari games?

### 4.1 Experimental Setting

**Network Architecture.** A CNN was used as the encoding module of VPN, and the transition module consists of one option-conditional convolution layer which uses different weights depending on the option followed by a few more convolution layers. We used a residual connection [10] from the previous abstract-state to the next abstract-state so that the transition module learns the change of the abstract-state. The outcome module is similar to the transition module except that it does not have a residual connection and two fully-connected layers are used to produce reward and discount. The value module consists of two fully-connected layers. The number of layers and hidden units vary depending on the domain. These details are described in the supplementary material.

**Implementation Details.** Our algorithm is based on asynchronous n-step Q-learning [20] where n is 10 and 16 threads are used. The target network is synchronized after every 10K steps. We used the Adam optimizer [14], and the best learning rate and its decay were chosen from $\{0.0001, 0.0002, 0.0005, 0.001\}$ and $\{0.98, 0.95, 0.9, 0.8\}$ respectively. The learning rate is multiplied by the decay every 1M steps. Our implementation is based on TensorFlow [1].[1]

VPN has four more hyperparameters: 1) the number of predictions steps (**k**) during training, 2) the plan depth (**$d_{train}$**) during training, 3) the plan depth (**$d_{test}$**) during evaluation, and 4) the branching factor (**b**) which indicates the number of options to be simulated for each expansion step during planning. We used $k = d_{train} = d_{test}$ throughout the experiment unless otherwise stated. **VPN(d)** represents our model which learns to predict and simulate up to d-step futures during training and evaluation. The branching factor ($b$) was set to 4 until depth of 3 and set to 1 after depth of 3, which means that VPN simulates 4-best options up to depth of 3 and only the best option after that.

**Baselines.** We compared our approach to the following baselines.

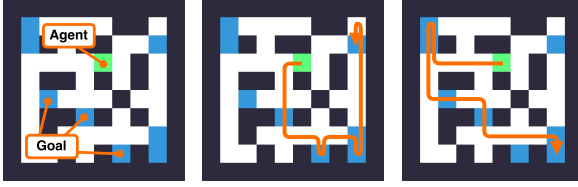 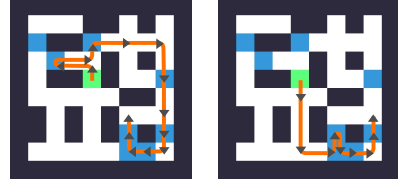

<div>

(a) Observation    (b) DQN's trajectory (c) VPN's trajectory

Figure 4: Collect domain. (a) The agent should collect as many goals as possible within a time limit which is given as additional input. (b-c) DQN collects 5 goals given 20 steps, while VPN(5) found the optimal trajectory via planning which collects 6 goals.

(a) Plan with 20 steps (b) Plan with 12 steps

Figure 5: Example of VPN's plan. VPN can plan the best future options just from the current state. The figures show VPN's different plans depending on the time limit.

</div>

- **DQN**: This baseline directly estimates Q-values as its output and is trained through asynchronous n-step Q-learning. Unlike the original DQN, however, our DQN baseline takes an option as additional input and applies an option-conditional convolution layer to the top of the last encoding convolution layer, which is very similar to our VPN architecture.[2]

- **VPN(1)**: This is identical to our VPN with the same training procedure except that it performs only 1-step rollout to estimate Q-value as shown in Figure 1a. This can be viewed as a variation of DQN that predicts reward, discount, and the value of the next state as a decomposition of Q-value.

- **OPN(d)**: We call this Observation Prediction Network (OPN), which is similar to VPN except that it directly predicts future observations. More specifically, we train two independent networks: a model network ($f^{model} : \mathbf{x}, \mathbf{o} \mapsto r, \gamma, \mathbf{x}'$) which predicts reward, discount, and the next observation, and a value network ($f^{value} : \mathbf{x} \mapsto V(\mathbf{x})$) which estimates the value from the observation. The training scheme is similar to our algorithm except that a squared loss for observation prediction is used to train the model network. This baseline performs d-step planning like VPN(d).

## 4.2 Collect Domain

**Task Description.** We defined a simple but challenging 2D navigation task where the agent should collect as many goals as possible within a time limit, as illustrated in Figure 4. In this task, the agent, goals, and walls are randomly placed for each episode. The agent has four options: move left/right/up/down to the first crossing branch or the end of the corridor in the chosen direction. The agent is given 20 steps for each episode and receives a positive reward (2.0) when it collects a goal by moving on top of it and a time-penalty ($-0.2$) for each step. Although it is easy to learn a sub-optimal policy which collects nearby goals, finding the optimal trajectory in each episode requires careful planning because the optimal solution cannot be computed in polynomial time.

An observation is represented as a 3D tensor ($\mathbb{R}^{3\times10\times10}$) with binary values indicating the presence/absence of each object type. The time remaining is normalized to $[0, 1]$ and is concatenated to the 3rd convolution layer of the network as a channel.

We evaluated all architectures first in a deterministic environment and then investigated the robustness in a stochastic environment separately. In the stochastic environment, each goal moves by one block with probability of 0.3 for each step. In addition, each option can be repeated multiple times with probability of 0.3. This makes it difficult to predict and plan the future precisely.

**Overall Performance.** The result is summarized in Figure 6. To understand the quality of different policies, we implemented a greedy algorithm which always collects the nearest goal first and a shortest-path algorithm which finds the optimal solution through exhaustive search assuming that the environment is deterministic. Note that even a small gap in terms of reward can be qualitatively substantial as indicated by the small gap between greedy and shortest-path algorithms.

The results show that many architectures learned a better-than-greedy policy in the deterministic and stochastic environments except that OPN baselines perform poorly in the stochastic environment. In addition, the performance of VPN is improved as the plan depth increases, which implies that deeper predictions are reliable enough to provide more accurate value estimates of future states. As a result, VPN with 5-step planning represented by 'VPN(5)' performs best in both environments.

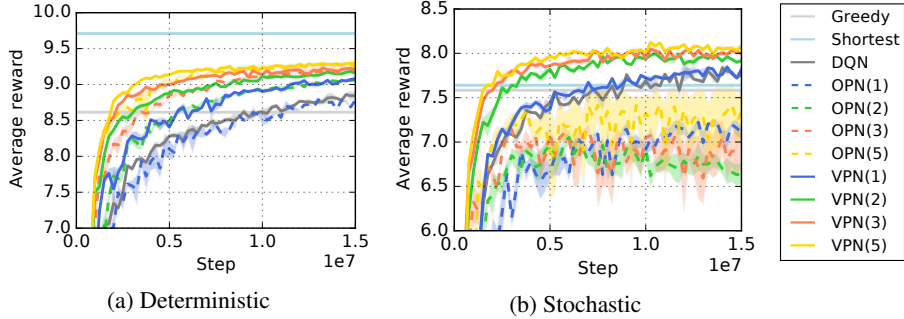

Figure 6: Learning curves on Collect domain. 'VPN(d)' represents VPN with d-step planning, while 'DQN' and 'OPN(d)' are the baselines.

**Comparison to Model-free Baselines.** Our VPNs outperform DQN and VPN(1) baselines by a large margin as shown in Figure 6. Figure 4 (b-c) shows an example of trajectories of DQN and VPN(5) given the same initial state. Although DQN's behavior is reasonable, it ended up with collecting one less goal compared to VPN(5). We hypothesize that 6 convolution layers used by DQN and VPN(1) are not expressive enough to find the best route in each episode because finding an optimal path requires a combinatorial search in this task. On the other hand, VPN can perform such a combinatorial search to some extent by simulating future abstract-states, which has advantages over model-free approaches for dealing with tasks that require careful planning.

**Comparison to Observation-based Planning.** Compared to OPNs which perform planning based on predicted observations, VPNs perform slightly better or equally well in the deterministic environment. We observed that OPNs can predict future observations very accurately because observations in this task are simple and the environment is deterministic. Nevertheless, VPNs learn faster than OPNs in most cases. We conjecture that it takes additional training steps for OPNs to learn to predict future observations. In contrast, VPNs learn to predict only minimal but sufficient information for planning: reward, discount, and the value of future abstract-states, which may be the reason why VPNs learn faster than OPNs.

In the stochastic Collect domain, VPNs significantly outperform OPNs. We observed that OPNs tend to predict the average of possible future observations ($\mathbb{E}_{\mathbf{x}}[\mathbf{x}]$) because OPN is deterministic. Estimating values on such blurry predictions leads to estimating $V_\theta(\mathbb{E}_{\mathbf{x}}[\mathbf{x}])$ which is different from the true expected value $\mathbb{E}_{\mathbf{x}}[V(\mathbf{x})]$. On the other hand, VPN is trained to approximate the true expected value because there is no explicit constraint or loss for the predicted abstract state. We hypothesize that this key distinction allows VPN to learn different modes of possible future states more flexibly in the abstract state space. This result suggests that a value-prediction model can be more beneficial than an observation-prediction model when the environment is stochastic and building an accurate observation-prediction model is difficult.

**Generalization Performance.** One advantage of model-based RL approach is that it can generalize well to unseen environments as long as the dynamics of the environment remains similar. To see if our VPN has such a property, we evaluated all architectures on two types of previously unseen environments with either reduced number of goals (from 8 to 5) or increased number of walls. It turns out that our VPN is much more robust to the unseen environments compared to model-free baselines (DQN and VPN(1)), as shown in Table 1. The model-free baselines perform worse than the greedy algorithm on unseen environments, whereas VPN still performs well. In addition, VPN generalizes as well as OPN which can learn a near-perfect model in the deterministic setting, and VPN significantly outperforms

Table 1: Generalization performance. Each number represents average reward. 'FGs' and 'MWs' represent unseen environments with fewer goals and more walls respectively. Bold-faced numbers represent the highest rewards with 95% confidence level.

|  | Deterministic | | | Stochastic | | |
|---|---|---|---|---|---|---|
|  | Original | FGs | MWs | Original | FGs | MWs |
| Greedy | 8.61 | 5.13 | 7.79 | 7.58 | 4.48 | 7.04 |
| Shortest | 9.71 | 5.82 | 8.98 | 7.64 | 4.36 | 7.22 |
| DQN | 8.66 | 4.57 | 7.08 | 7.85 | 4.11 | 6.72 |
| VPN(1) | 8.94 | 4.92 | 7.64 | 7.84 | 4.27 | 7.15 |
| OPN(5) | **9.30** | **5.45** | **8.36** | 7.55 | 4.09 | 6.79 |
| **VPN(5)** | **9.29** | **5.43** | **8.31** | **8.11** | **4.45** | **7.46** |

OPN in the stochastic setting. This suggests that VPN has a good generalization property like model-based RL methods and is robust to stochasticity.

Table 2: Performance on Atari games. Each number represents average score over 5 top agents.

|  | Frostbite | Seaquest | Enduro | Alien | Q*Bert | Ms. Pacman | Amidar | Krull | Crazy Climber |
|---|---|---|---|---|---|---|---|---|---|
| DQN | 3058 | 2951 | 326 | **1804** | 12592 | **2804** | 535 | 12438 | 41658 |
| **VPN** | **3811** | **5628** | **382** | 1429 | **14517** | 2689 | **641** | **15930** | **54119** |

**Effect of Planning Depth.** To further investigate the effect of planning depth in a VPN, we measured the average reward in the deterministic environment by varying the planning depth ($d_{test}$) from 1 to 10 during evaluation after training VPN with a fixed number of prediction steps and planning depth ($k, d_{train}$), as shown in Figure 7. Since VPN does not learn to predict observations, there is no guarantee that it can perform deeper planning during evaluation ($d_{test}$) than the planning depth used during training ($d_{train}$). Interestingly, however, the result in Figure 7 shows that if $k = d_{train} > 2$, VPN achieves better performance during evaluation through deeper tree search ($d_{test} > d_{train}$). We also tested a VPN with $k = 10$ and $d_{train} = 5$ and found that a planning depth of 10 achieved the best performance

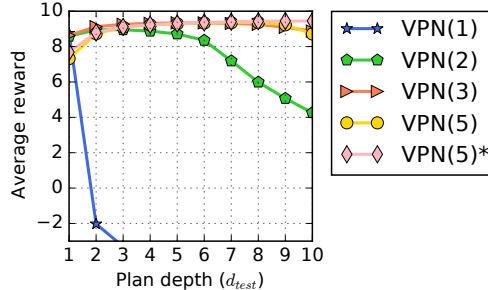

Figure 7: Effect of evaluation planning depth. Each curve shows average reward as a function of planning depth, $d_{test}$, for each architecture that is trained with a fixed number of prediction steps. 'VPN(5)*' was trained to make 10-step predictions but performed 5-step planning during training ($k = 10, d_{train} = 5$).

during evaluation. Thus, with a suitably large number of prediction steps during training, our VPN is able to benefit from deeper planning during evaluation relative to the planning depth during training. Figure 5 shows examples of good plans of length greater than 5 found by a VPN trained with planning depth 5. Another observation from Figure 7 is that the performance of planning depth of 1 ($d_{test} = 1$) degrades as the planning depth during training ($d_{train}$) increases. This means that a VPN can improve its value estimations through long-term planning at the expense of the quality of short-term planning.

## 4.3 Atari Games

To investigate how VPN deals with complex visual observations, we evaluated it on several Atari games [2]. Unlike in the Collect domain, in Atari games most primitive actions have only small value consequences and it is difficult to hand-design useful extended options. Nevertheless, we explored if VPNs are useful in Atari games even with short-lookahead planning using simple options that repeat the same primitive action over extended time periods by using a frame-skip of 10.[3] We pre-processed the game screen to $84 \times 84$ gray-scale images. All architectures take last 4 frames as input. We doubled the number of hidden units of the fully-connected layer for DQN to approximately match the number of parameters. VPN learns to predict rewards and values but not discount (since it is fixed), and was trained to make 3-option-step predictions for planning which means that the agent predicts up to 0.5 seconds ahead in real-time.

As summarized in Table 2 and Figure 8, our VPN outperforms DQN baseline on 7 out of 9 Atari games and learned significantly faster than DQN on Seaquest, QBert, Krull, and Crazy Climber. One possible reason why VPN outperforms DQN is that even 3-step planning is indeed helpful for learning a better policy. Figure 9 shows an example of VPN's 3-step planning in Seaquest. Our VPN predicts reasonable values given different sequences of actions, which can potentially help choose a better action by looking at the short-term future. Another hypothesis is that the architecture of VPN itself, which has several auxiliary prediction tasks for multi-step future rewards and values, is useful for learning a good abstract-state representation as a model-free agent. Finally, our algorithm which performs planning to compute the target Q-value can potentially speed up learning by generating more accurate targets as it performs value backups multiple times from the simulated futures, as discussed in Section 3.4. These results show that our approach is applicable to complex visual environments without needing to predict observations.

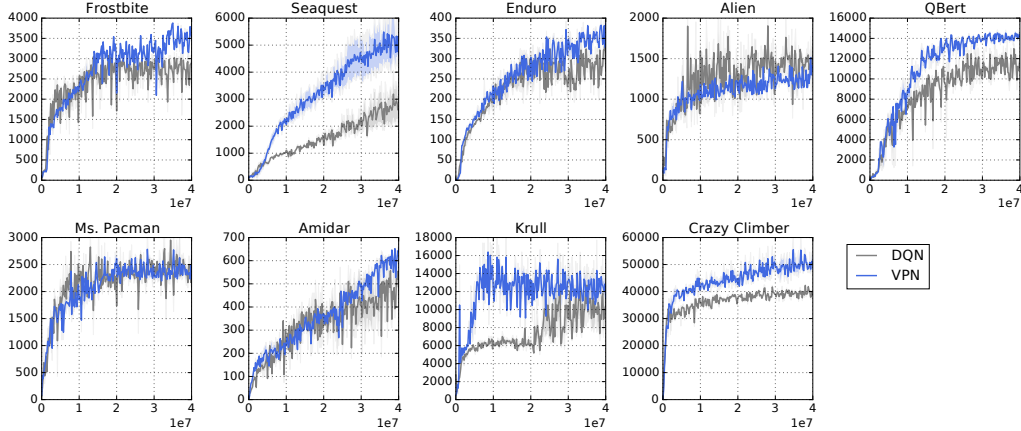

Figure 8: Learning curves on Atari games. X-axis and y-axis correspond to steps and average reward over 100 episodes respectively.



(a) State      (b) Plan 1 (19.3)      (c) Plan 2 (18.7)      (d) Plan 3 (18.4)      (e) Plan 4 (17.1)

Figure 9: Examples of VPN's value estimates. Each figure shows trajectories of different sequences of actions from the initial state (a) along with VPN's value estimates in the parentheses: $r_1 + \gamma r_2 + \gamma^2 r_3 + \gamma^3 V(s_4)$. The action sequences are (b) DownRight-DownRightFire-RightFire, (c) Up-Up-Up, (d) Left-Left-Left, and (e) Up-Right-Right. VPN predicts the highest value for (b) where the agent kills the enemy and the lowest value for (e) where the agent is killed by the enemy.

## 5 Conclusion

We introduced value prediction networks (VPNs) as a new deep RL way of integrating planning and learning while simultaneously learning the dynamics of abstract-states that make option-conditional predictions of future rewards/discount/values rather than future observations. Our empirical evaluations showed that VPNs outperform model-free DQN baselines in multiple domains, and outperform traditional observation-based planning in a stochastic domain. An interesting future direction would be to develop methods that automatically learn the options that allow good planning in VPNs.

## Acknowledgement

This work was supported by NSF grant IIS-1526059. Any opinions, findings, conclusions, or recommendations expressed here are those of the authors and do not necessarily reflect the views of the sponsor.

## Footnotes

[1]The code is available on `https://github.com/junhyukoh/value-prediction-network`.

[2]This architecture outperformed the original DQN architecture in our preliminary experiments.

[3]Much of the previous work on Atari games has used a frame-skip of 4. Though using a larger frame-skip generally makes training easier, it may make training harder in some games if they require more fine-grained control [17].

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
