[Supplementary Material · supplementary.pdf]

# Supplementary Material: Value Prediction Network

**Junhyuk Oh**[†]     **Satinder Singh**[†]     **Honglak Lee**[∗,†]

[†]University of Michigan
[∗]Google Brain
{junhyuk,baveja,honglak}@umich.edu, honglak@google.com

## A    Comparison between VPN and DQN in the Deterministic Collect

Figure 1: Examples of trajectories and planning on the deterministic Collect domain. The first column shows initial observations, and the following two columns show trajectories of DQN and VPN respectively. It is shown that DQN sometimes chooses a non-optimal option and ends up with collecting fewer goals than VPN. The last column visualizes VPN's 10 option-step planning from the initial state. Note that VPN's initial plans do not always match with its actual trajectories because the VPN re-plans at every step as it observes its new state.

# B Comparison between VPN and OPN in the Stochastic Collect

Figure 2: Example of trajectory on the stochastic Collect domain. Each row shows a trajectory of VPN (top) and OPN (bottom) given the same initial state. At t=6, the VPN decides to move up to collect nearby goals, while the OPN moves left to collect the other goals. As a result, the OPN collects two fewer goals compared to VPN. Since goals move randomly and the outcome of option is stochastic, the agent should take into account many different possible futures to find the best option with the highest expected outcome. Though the outcome is noisy due to the stochasticity of the environment, our VPN tends to make better decisions more often than OPN does in expectation.

# C   Examples of Planning on Atari Games

Figure 3: Examples of VPN's planning on Atari games. The first column shows initial states, and the following columns show our VPN's value estimates in parentheses given different sequences of actions. Red and black arrows represent movement actions with and without 'Fire'. 'N' and 'F' correspond to 'No-operation' and 'Fire'. (Seaquest) The VPN estimates higher values for moving up to refill the oxygen tank and lower values for moving down to kill enemies because the agent loses a life when the oxygen tank is empty, which is almost running out. (Ms. Pacman) The VPN estimates the lowest value for moving towards an enemy (ghost). It also estimates a low value for moving right because it already has eaten some yellow pellets on the right side. On the other hand, it estimates relatively higher values for moving left and down because it collects more pellets while avoiding the enemy. (Frostbite) The VPN estimates higher values for collecting a nearby fish, which gives a positive reward, and lower values for not collecting it. (Enduro) The VPN estimates higher values for accelerating (fire) and avoiding collision and lower values for colliding with other cars.

# D   Details of Learning

Algorithm 1 describes our algorithm for training value prediction network (VPN). We observed that training the outcome module (reward and discount prediction) on additional data collected from a random policy slightly improves the performance because it reduces a bias towards the agent's behavior. More specifically, we fill a replay memory with $R$ transitions from a random policy before training and sample transitions from the replay memory to train the outcome module. This procedure is described in Line 4 and Lines 20-24 in Algorithm 1. This method was used only for Collect domain (not for Atari) in our experiment by generating 1M transitions from a random policy.

---

**Algorithm 1** Asynchronous n-step Q-learning with k-step prediction and d-step planning

---

1: $\theta$: *global parameter, $\theta^-$: global target network parameter, $T$: global step counter*
2: *$d$: plan depth, $k$: number of prediction steps*
3: $t \leftarrow 0$ and $T \leftarrow 0$
4: $M[1...R] \leftarrow$ Store $R$ transitions $(s, o, r, \gamma, s')$ using a random policy
5: **while** not converged **do**
6:     Clear gradients $d\theta \leftarrow 0$
7:     Synchronize thread-specific parameter $\theta' \leftarrow \theta$
8:     $t_{start} \leftarrow t$
9:     $s_t \leftarrow$ Observe state
10:     **while** $t - t_{start} < n$ and $s_t$ is non-terminal **do**
11:         $a_t \leftarrow \text{argmax}_o Q_{\theta'}^d(s_t, o_t)$ or random option based on $\epsilon$-greedy policy
12:         $r_t, \gamma_t, s_{t+1} \leftarrow$ Execute $o_t$
13:         $t \leftarrow t + 1$ and $T \leftarrow T + 1$
14:     **end while**
15:     $R = \begin{cases} 0 & \text{if } s_t \text{ is terminal} \\ \max_o Q_{\theta^-}^d(s_t, o) & \text{if } s_t \text{ is non-terminal} \end{cases}$
16:     **for** $i = t - 1$ to $t_{start}$ **do**
17:         $R \leftarrow r_i + \gamma_i R$
18:         $d\theta \leftarrow d\theta + \nabla_{\theta'} \left[ \sum_{l=1}^k \left( R - v_i^l \right)^2 + \left( r_i - r_i^l \right)^2 + \left( \log_\gamma \gamma_i - \log_\gamma \gamma_i^l \right)^2 \right]$
19:     **end for**
20:     $t' \leftarrow$ Sample an index from $1, 2, ..., R$
21:     **for** $i = t'$ to $t' + n$ **do**
22:         $s_i, a_i, r_i, \gamma_i, s_{i+1} \leftarrow$ Retrieve a transition from $M[i]$
23:         $d\theta \leftarrow d\theta + \nabla_{\theta'} \left[ \sum_{l=1}^k \left( r_i - r_i^l \right)^2 + \left( \log_\gamma \gamma_i - \log_\gamma \gamma_i^l \right)^2 \right]$
24:     **end for**
25:     Perform asynchronous update of $\theta$ using $d\theta$
26:     **if** $T$ mod $I_{target} == 0$ **then**
27:         Update the target network $\theta^- \leftarrow \theta$
28:     **end if**
29: **end while**

---

# E   Details of Hyperparameters

Figure 4: Transition module used for Collect domain. The first convolution layer uses different weights depending on the given option. Sigmoid activation function is used for the last 1x1 convolution such that its output forms a mask. This mask is multiplied to the output from the 3rd convolution layer. Note that there is a residual connection from **s** to **s′**. Thus, the transition module learns the change of the consecutive abstract states.

## E.1   Collect

The encoding module of our VPN consists of Conv(32-3x3-1)-Conv(32-3x3-1)-Conv(64-4x4-2) where Conv(N-KxK-S) represents N filters with size of KxK with a stride of S. The transition module is illustrated in Figure 4. It consists of OptionConv(64-3x3-1)-Conv(64-3x3-1)-Conv(64-3x3-1) and a separate Conv(64-1x1-1) for the mask which is multiplied to the output of the 3rd convolution layer of the transition module. 'OptionConv' uses different convolution weights depending on the given option. We also used a residual connection from the previous abstract state to the next abstract state such that the transition module learns the difference between two states. The outcome module has OptionConv(64-3x3-1)-Conv(64-3x3-1)-FC(64)-FC(2) where FC(N) represents a fully-connected layer with N hidden units. The value module consists of FC(64)-FC(1). Exponential linear unit (ELU) [1] was used as an activation function for all architectures.

Our DQN baseline consists of the encoding module followed by the transition module followed by the value module. Thus, the overall architecture is very similar to VPN except that it does not have the outcome module. To match the number of parameters, we used 256 hidden units for DQN's value module. We found that this architecture outperforms the original DQN architecture [2] on Collect domain and several Atari games.

The model network of OPN baseline has the same architecture as VPN except that it has an additional *decoding* module which consists of Deconv(64-4x4-2)-Deconv(32-3x3-1)-Deconv(32-3x3-1). This module is applied to the predicted abstract-state so that it can predict the future observations. The value network of OPN has the same architecture as our DQN baseline.

A discount factor of $0.98$ was used, and the target network was synchronized after every 10K steps. The epsilon for $\epsilon$-greedy policy was linearly decreased from $1$ to $0.05$ for the first 1M steps.

## E.2   Atari Games

The encoding module consists of Conv(16-8x8-4)-Conv(32-4x4-2), and the transition module has OptionConv(32-3x3-1)-Conv(32-3x3-1) with a mask and a residual connection as described above. The outcome module has OptionConv(32-3x3-1)-Conv(32-3x3-1)-FC(128)-FC(1), and the value module consists of FC(128)-FC(1). The DQN baseline has the same encoding module followed by the transition module and the value module, and we used 256 hidden units for the value module of DQN to approximately match the number of parameters. The other hyperparameters are same as the ones used in the Collect domain except that a discount factor of $0.99$ was used.