[Reviews · NeurIPS 2017]

Reviewer 1



This paper propose a reinforcement learning planning algorithm, called VPN (value prediction network), that works for discrete action, continuous state problems. The framework is similar to the Dyna architecture, including a value prediction model, a learned transient model, and a planning module that do single step or multiple steps of rollouts. Experiments were performed on a collect domain and several Atari games. The paper cites Dyna, but the more relevant is the linear Dyna work: Single step liner Dyna: http://uai2008.cs.helsinki.fi/UAI_camera_ready/sutton.pdf Multi-step linear Dyna: https://papers.nips.cc/paper/3670-multi-step-dyna-planning-for-policy-evaluation-and-control.pdf Both papers, though, does not deal with options. The paper seems to emphasize the difference of VPN to a similar algorithm called OPN. In VPN, the value is a function of the abstract state, but not the original observation. I agree this is true. Then in OPN, as discussed from line 200, the value is a function of the observation. I feel making such distinction is unnecessary. At the merit of making such a distinction is unclear. Taking Dyna and linear Dyna for example, Dyna uses a model between original observations (state is finite); linear Dyna uses a model between abstract models (continuous state problems). There is no agreement that Dyna or linear Dyna is better. Instead, they work for different problems. Making a comparison between models built on original observations or abstract state may depend on algorithmic implementation. Figure 7: planning depth study is interesting. Some intermediate, usually small number of planning steps performs the best. This is consistent with Yao’s results. In Figure 2 of the linked paper, 10 planning steps (instead of only 1 or infinite planning steps) performed the best. The option used in Ataris game are simple repeating primitive actions. This might be too simple except for algorithmic study convenience.

Reviewer 2



This paper proposes a new learning architecture, VPN, whose components can be learned end-to-end to perform planning over pre-specified temporally extended actions. An advantage of this approach is that it explicitly avoids working in the original observation space and plans indirectly over a learned representation. This is made possible by a applying a model-free DQN-like approach for learning values which are then used in combination with reward and duration models of options to form planning targets. The proposed approach is shown to outperform DQN in a variety of Atari games. Model-based RL is a difficult problem when it comes to a large domain like ALE. The development of new model-based methods which do not explicitly try to reconstruct/predict the next observation seem to be going in the right direction. VPN is a contribution in this new line of research. However, there are a few conceptual elements which remain unclear. First, why predicting the expected discount/duration separately from the option-conditional expected cumulative return to termination ? Using the Bellman-like equations for the reward model of options (see equation 17 of the options paper), the discount factor can effectively be folded into the reward prediction. This would avoid the need to learn a separate discount prediction for each option. Second : The nature of the "next abstract state" (line 109) is a key element of the proposed approach to avoid learning a transition model directly in the original space. As you mention later on line 251, there is no incentive/constraint for the learned state representation to capture useful properties for planning and could in the extreme case become a tabular representation. The appendix then shows that you use a particular architecture for the transition module which predicts the difference. How much this prior on the architecture affects quality of the representation learned by the system ? What kind of representation would be learned in the absence of a residual connection ? Can you also elaborate on the meaning of the weighting (1/d, d-1/d) in equation (1) ? Why is it necessary ? Where does that come from ? Line 21 : "predict abstractions of observations" This expression is unclear, but I guess that it means a feature/representation of the state. You also write : "not clear how to roll such predictions", but isn't that exactly what VPN is doing ? Typo : line 31 "dyanmics"